# Distinct Effects of GnRH Immunocastration Versus Surgical Castration on Gut Microbiota

**DOI:** 10.3390/ani15243512

**Published:** 2025-12-05

**Authors:** Fanli Kong, Ruohan Yang, Xingyu Zhou, Yuanyuan Shen, Wenhao Wei, Xianyin Zeng, Xiaogang Du, Xinfa Han

**Affiliations:** 1College of Life Science, Sichuan Agricultural University, Ya’an 625014, China; yrh18382031118@163.com (R.Y.); weiwenhao@stu.sicau.edu.cn (W.W.); yzeng@sicau.edu.cn (X.Z.); duxiaogang@sicau.edu.cn (X.D.); xfhan2012@163.com (X.H.); 2Livestock and Poultry Multi-Omics Key Laboratory of Ministry of Agriculture and Rural Affairs, College of Animal Science and Technology, Sichuan Agricultural University, Chengdu 611130, China; 202400457@stu.sicau.edu.cn (X.Z.); 202300259@stu.sicau.edu.cn (Y.S.)

**Keywords:** gut microbiota, surgical castration, GnRH immunocastration, SD rats, sexual dimorphism

## Abstract

Surgical castration raises animal welfare issues and can affect growth. Active immunization against gonadotropin-releasing hormone (GnRH immunocastration) is a kinder alternative that also suppresses sex hormones. This study in SD rats compared the effects of these two methods on gut microbiota. Both effectively suppressed gonadal development, but they had different impacts on the gut microbiota. Surgical castration reduced the gut microbial diversity, but GnRH immunocastration kept it closer to the control group. Both methods changed specific genera, but in some cases, the changes were opposite. The predicted microbial functions were also altered differently in female and male SD rats. In conclusion, GnRH immunocastration is less disruptive to the gut microbial community. This suggests it is a potentially less stressful alternative that better preserves a stable gut environment, which is an important aspect of animal health.

## 1. Introduction

Surgical castration, the physical removal of reproductive organs, is a common practice for controlling animal reproduction, mitigating aggressive and sexual behaviors, particularly in pet management and livestock farming. In livestock farming, surgical castration can prevent boar taint to improve meat flavor, promote fat deposition, enhance meat tenderness, to increase market value [1]. However, this procedure raises significant animal welfare concerns, e.g., enduring pain and experiencing prolonged stress responses, which can adversely affect production efficiency, such as slower growth rates, less efficient feed conversion, and increased fat deposition [2,3]. These drawbacks have driven the search for milder alternatives that achieve the benefits of castration while minimizing harm to the animal.

Active immunization against gonadotropin-releasing hormone (GnRH) castration has emerged as a promising animal-friendly alternative, suppressing the synthesis and secretion of sex hormones without surgery [4,5,6,7]. Specifically, it reduces endogenous GnRH levels, thereby significantly inhibiting the synthesis and secretion of luteinizing hormone (LH) and follicle-stimulating hormone (FSH), which are crucial for reproductive functions [8]. In our previous work, Han et al. reported a continuous decline in testosterone, LH, and FSH levels after GnRH immunization in rats, reaching nearly undetectable levels within two weeks, indicating the suppression of the reproductive axis [9,10,11,12]. Meanwhile, the long-term suppression of reproductive function after GnRH immunocastration has been demonstrated in various species, including adult cats, pigs, and sheep [7,13,14]. Moreover, Zeng et al. found that GnRH-immunized boars outperformed surgically castrated pigs, exhibiting faster growth, improved feed utilization, and leaner carcasses [15]. These findings indicated that GnRH immunocastration can achieve husbandry outcomes comparable to surgical castration.

Currently, extensive studies have reported the impacts of GnRH immunocastration on reproductive hormone levels and functionality of reproductive organs [16,17]. While both surgical and GnRH immunocastration ultimately reduce circulating sex hormone levels, their mechanisms of action are fundamentally different [18]. The gut microbiome, a complex community of microorganisms residing in the intestines, plays a pivotal role in host digestion, immunity, and metabolism [19]. More evidence from human and animal studies suggested that sex hormones are a key factor shaping gut microbial composition [20,21,22]. Especially, Markle et al. demonstrated that the gut microbiota of castrated male mice shifts toward a female-like composition, indicating that androgens are a major determinant of sex-specific microbial profiles [21]. Whether these distinct interventions exert differential effects on the gut microbiota is still relatively limited.

The Sprague Dawley (SD) rat is an excellent model for biomedical research due to its disease resistance, low tumor incidence, and high sensitivity to sex hormones. In this study, we aimed to comparatively explore the effects of GnRH immunocastration versus surgical castration on the gut microbiota in SD rats. By identifying gut microbiota features associated with each castration method, our work not only advances the understanding of castration-specific impacts but also contributes to the broader knowledge of the interplay between sex hormones and the gut ecosystem.

## 2. Materials and Methods

### 2.1. Ethics Statement

All animal experiments were conducted in strict accordance with the Regulations for the Administration of Affairs Concerning Experimental Animals (Ministry of Science and Technology, China, revised in March 2017). The experimental protocol was reviewed and approved by the animal ethics and welfare committee (AEWC) of Sichuan Agricultural University under permit No. 20240621.

### 2.2. Experimental Animals and Design

Sixty specific pathogen-free (SPF) Sprague Dawley (SD) rats (aged 4–5 weeks) with similar body weight and good health were purchased from Chengdu Dashuo Experimental Animal Co., Ltd., Chengdu, China. All animals were confirmed by the vendor to be free from a standard set of pathogens, including viruses, bacteria, and parasites.

After one week of acclimatization, SD rats were randomly assigned to three groups (*n* = 20 per group; 10 males and 10 females): (1) Control group: This group served as the baseline control and did not undergo any surgical or immunological procedures throughout the entire experiment. (2) Surgical castration group: Rats in this group underwent bilateral ovariectomy (female) or orchiectomy (male) under ether anesthetized at 7 weeks of age. (3) GnRH immunocastration group: Rats in this group received primary immunization at 9 weeks of age and a booster immunization at 17 weeks of age via intramuscular injection in the hind leg with a GnRH vaccine (Figure 1). All rats were housed under standard conditions (temperature 23–25 °C, humidity 50–60%, 12 h light/dark cycle) with ad libitum access to food and water.

The GnRH vaccine was prepared as previously described by Oonk et al. and our previous work by Han et al. [9,10,23]. Briefly, a GnRH juxtaposition dimer using D-lysine to replace glycine at position 6 and an ovalbumin linker (TDK-OVA) were dissolved in PBS (pH = 7.4) and fully emulsified with an equal volume of Specol adjuvant. The injection dose (50 mg GnRH peptide equivalent of conjugate) and method were the same between primary and booster immuniocastration. All animals were sacrificed at 23 weeks of age.

### 2.3. Sample Collection

Fresh fecal samples (200 mg) from each SD rat in all groups were collected at three age points corresponding to 8 days post-surgical castration (8 weeks of age), 35 days after primary immunocastration (14 weeks of age), and 27 days after booster immunocastration (21 weeks of age). Given the distinct castration procedure modes, we used chronological age as the sampling anchor. A total of 180 fecal samples were collected, immediately frozen in liquid nitrogen, and stored at −80 °C for subsequent DNA extraction.

At 23 weeks of age, all SD rats were euthanized. Bilateral ovaries and testes were harvested from rats in the control and GnRH immunocastration groups and weighed after removing adhering adipose tissue.

### 2.4. Microbial DNA Extraction and 16S rRNA Gene Sequencing

Microbial genomic DNA was extracted from each fecal sample using the OMEGA Stool DNA Kit (M5635-01, Omega Bio-Tek, Norcross, GA, USA), according to the manufacturer’s instructions. The extracted DNA was treated with RNase to eliminate RNA contamination. DNA concentration and integrity were assessed using a Nanodrop1000 (Thermo Fisher Scientific, Waltham, MA, USA) and by 0.8% agarose gel electrophoresis.

The hypervariable V3–V4 regions of the microbial 16S rRNA gene were amplified using the primer pairs (338F: 5′-ACTCCTACGGGAGGCAGCA-3′, 806R: 5′-GGACTACHVGGGTWTCTAAT-3′). Amplification products were used to construct sequencing libraries, which were sequenced on an Illumina NovaSeq 6000 platform (Novogene, Beijing, China).

### 2.5. 16S rRNA Gene Sequencing Data Processing and Analysis

Raw sequencing reads were processed using the QIIME2 pipeline (version 2021.11) according to a previously suggested workflow [24]. Briefly, the reads were trimmed, dereplicated, and combined. The chimeras and low-quality sequences (relative abundance of ASV < 0.005%) were removed using vsearch [25]. Then, the resulting clean reads were clustered into amplicon sequence variants (ASV) based on 99% sequence similarity. Taxonomic assignment of ASVs was performed against the Silva database (v138) [26].

To investigate the variation in microbial communities across samples, the Shannon index and observed features index for α diversity were calculated. Bray–Curtis and Jaccard distances for β diversity analysis were calculated and visualized by principal coordinates analysis (PCoA).

Putative metagenomic functions were predicted from 16S rRNA data using PICRUSt2 [27].

### 2.6. Statistical Analysis

To analyze taxonomic differences in microbial abundance, we used linear discriminant analysis (LDA) effect size (LEfSe) (http://huttenhower.sph.harvard.edu/galaxy/, accessed on 2 December 2025) [28].

To identify the most important microbial metabolic pathways for discriminating among each group, we performed a machine learning-based analysis using the Random Forest classifier, which was implemented using the randomForest package (v4.7-1.2) in R (v4.5.1), with parameters set to 1000 decision trees (ntree = 1000). The importance of each pathway was evaluated based on the “Mean Decrease Accuracy” metric.

Statistical comparisons were analyzed using pairwise Wilcoxon tests. Pairwise permutational multivariate analysis of variance (PERMANOVA) with 999 random permutations was performed to test the significance of differences among groups.

A *p* value less than 0.05 was considered statistically significant.

## 3. Results

### 3.1. GnRH Immunocastration Reduced Gonadal Weight in SD Rats

At the end of the experiment (23 weeks of age), SD rats were euthanized, and the ovaries and testes were collected and weighed. As shown in Figure 2, GnRH immunocastration resulted in a significant reduction in gonad weight compared to their respective sex-matched control group in female and male SD rats. Specifically, the ovarian weight of female SD rats in the immunocastration group was significantly lower than that in the female control group (*p* < 0.05) (Figure 2A). Similarly, the testicular weight of male SD rats in the immunocastration group was significantly lower than that in the male control group (*p* < 0.05) (Figure 2B). These results clearly demonstrated that GnRH immunocastration effectively suppressed gonadal development in SD rats [6,7].

### 3.2. General Characteristics of Gut Microbiota in SD Rats

A total of 11,469,744 valid sequences was obtained from 180 fecal samples, with an average of 63,721 sequences per sample. After data trimming and quality filtering, 4,728,829 high-quality sequences were retained, with an average of 26,271 sequences per sample (ranging from 13,617 to 53,065). The high-quality sequences were clustered into 970 amplicon sequence variants (ASVs) based on the 99% sequence identity, which were taxonomically classified into 11 phyla and 105 genera. In general, gut microbiota in SD rats were predominantly composed of the Firmicutes and Bacteroidetes across all experimental groups (Figure 3A). At the genus level, the most abundant taxa included *Lactobacillus*, *Prevotella*, *Muribaculaceae*, *Bacteroides*, *Lachnospiraceae*_NK4A136_group, *Clostridia*_UCG-014, *Ruminococcus*, *Alistipes*, and *Prevotellaceae*_Ga6A1_group (Figure 3B). Notably, both surgical and immunological castration led to shifts in the microbial profile compared to the control groups and a visible difference in microbial composition between male and female rats within the same treatment group, suggesting a moderating effect of sex on the gut microbiota.

### 3.3. Surgical Castration Altered Gut Microbial Diversity and Composition in SD Rats

We then investigated the impact of surgical castration on gut microbiota, respectively, in female and male SD rats. The results showed microbial community alpha diversity based on the Shannon index and the number of observed features were significantly decreased in surgical castration SD rats (*p* <0.05) (Figure 4A,B). However, the effect of surgical castration on microbial community composition and structure was sex-dependent. In general, the community structure based on Bray–Curtis distances in SD rats was less affected, with a considerable overlap between surgical and control groups in the PCoA plot (Figure 4C,D). The community composition based on Jaccard distances both showed clear separations between surgical castration and control groups in both female and male SD rats (*p* < 0.001) (Figure 4E,F). The LEfSe analysis was further employed to identify bacterial taxa differentially abundant between the surgical castration and control groups. In female rats, the surgical castration group was characterized by a higher abundance of *Desulfovibrio* and a lower abundance of *Gastranaerophilales* (Figure 4G), whereas in male surgical castration rats, the group was significantly enriched in *Lachnospira* and reduced *Parasutterella* (Figure 4H). In addition, a higher abundance of *Butyricicoccaceae*_UCG008 and a lower abundance of *Butyricicoccus*, *Candidatus_Saccharimonas*, and *Candidatus_Arthromitus* were detected in both male and female surgical SD rats. These results indicated that surgical castration significantly altered the gut microbiota in SD rats, with the effect exhibiting a sexual dimorphism.

### 3.4. GnRH Immunocastration Modulates the Gut Microbiota in a Time and Sex-Dependent Manner

Next, to determine the effects of GnRH immunocastration on gut microbiota, we compared the gut microbiota of immunocastration groups with their corresponding age-matched controls after primary (14 weeks of age) and booster (21 weeks of age) GnRH immunocastration. The Shannon index was significantly reduced in gut microbial community diversity in male SD rats after booster immunocastration, but the diversity was similar to control groups in female SD rats (Figure 5A). No significant differences in the observed features index were detected either in female or in male SD rats after primary immunocastration and booster immunocastration (Figure 5B). Totally, principal coordinate analysis (PCoA) based on Bray–Curtis and Jaccard distances revealed that the microbial community structure and composition shifted with age over the course of the experiment in all groups of both female and male SD rats (Figure 5C,D). As expected, few gut microbial biomarker after immunocastration, especially the primary immunization, was detected using LEfSe analysis (Figure 5E). In addition, we also explored the dynamics trends of taxa after primary and booster immunocastration in SD rats (Appendix A). *Bacteroides*, *Alloprevotella*, *Parasutterella*, *Parabacteroides*, *Butyricimonas*, *Sutterella*, *Mycoplasma*, and *Candidatus_Arthromitus* were significantly reduced during primary and booster immunocastrations in both female and male SD rats, while *Clostridia*_UCG014, *Monoglobus*, and *Lactobacillus* were increased. A notable sex difference was observed for *Prevotella*, which was consistently reduced in females but exhibited a transient increase after primary immunocastration before declining in males (Figure 5F). These results demonstrated that the gut microbiota were altered after immunocastration, particularly after booster immunocastration, and the abundance of specific genera in a sexually dimorphic manner.

### 3.5. Comparative Analysis of Gut Microbiota Alterations Induced by GnRH Immunocastration and Surgical Castration in SD Rats

Furthermore, we compared the effects of gut microbiota between immunocastration and surgical castration (21 weeks of age). Interestingly, the gut microbial community diversity based on the Shannon index in immunocastration was higher than that of the surgical castration group in both female and male SD rats, especially significantly higher in male SD rats (Figure 6A). The community richness based on observed features in the immunocastration group was significantly higher than the surgical castration group in male group. Accordingly, principal coordinate analysis (PCoA) based on Bray–Curtis and Jaccard revealed a sexually dimorphic pattern (Figure 6B). Generally, the microbial community structure and community composition in female SD rats showed considerable overlap between immunocastration and surgical castration groups, whereas a marked separation was observed between the groups in male SD rats.

Then, to further compare microbial taxa that serve as biomarkers for immunocastration and surgical castration treatment in SD rats, we performed LEfSe analysis in female and male SD rats gut microbiota, respectively. In female SD rats, six genera, including *Romboutsia*, *Akkermansia*, *Oscillospiraceae*_NK4A214_group, *Rikenella*, *Treponema* and *Clostridia_vadin*BB60_group, were significantly different between immunocastration and surgical castration groups (Figure 6C). In male SD rats, a substantial difference was observed, with 30 genera serving as discriminative biomarkers, such as *Prevotella*, *Oscillospiraceae*_UCG005, *Bifidobacterium*, *Bacteroides*, *Christensenellaceae*_R7_group, *Prevotellaceae*_Ga6A1_group, and *Lactobacillus* (Figure 6D). *Oscillospiraceae*_NK4A214_group was enriched in the immunocatration group in both females and males. We also explored the gut microbiota that changed with the two castration treatments (Appendix A). *Clostridia*_UCG014, *Lactobacillus* and *Lachnospiraceae*_UCG006 were simultaneously changed in both immunocastration and surgical castration of female SD rats. *Intestinimonas* and *Erysipelatoclostridiaceae*_UCG004 were simultaneously changed in both immunocastration and surgical castration male SD rats (Figure 6E). Conversely, some taxa exhibited opposite responses; for example, *Eubacterium_nodatum*_group was significantly increased in GnRH immunocastration but decreased in surgical castration of male SD rats (Figure 6E). In conclusion, GnRH active immunization and surgical castration both perturbed the gut microbiota, but their effects are distinct.

At last, we used a computational tool, PICRUSt (Phylogenetic Investigation of Communities by Reconstruction of Unobserved States) [27], to explore metabolic potentials of GnRH immunocastration and surgical castration on gut microbiota in SD rats based on the 16S rRNA sequencing data. To identify the key microbial functional characteristics that best distinguish between immunocastration and surgical castration treatments, we performed Random Forest analysis. The results revealed a striking sexual dimorphism in the predicted microbiota functions. In females, the pathways involved in nitrogen metabolism, glyoxylate and dicarboxylate metabolism, mismatch repair, and biosynthesis of various cellular components (e.g., peptidoglycan, terpenoid backbones) best distinguished the microbial functions for immunocastration and surgical castration treatments in females (Figure 7A and Appendix A). It suggested that immunocastration and surgical castration treatment in female SD rats may differentially affect microbial core energy metabolism, immune and inflammatory responses, as well as the microbial adaptation mechanisms to the host’s internal environment. In contrast, the pathways encompassed key biological processes such as siderophore acquisition, energy metabolism, genetic information processing (e.g., DNA replication, RNA transport), amino acid metabolism, and pathogen interaction were discriminative of immunocastration and surgical castration treatments in male SD rats (Figure 7B and Appendix A). This indicated that the GnRH immunocastration and surgical castration treatments may induce distinct and sex-specific metabolic shifts in SD rats’ gut microbiota.

## 4. Discussion

The primary goal of this study was to comparatively investigate the effects of GnRH immunocastration and surgical castration on gut microbiota. Our findings clearly demonstrated that surgical castration and GnRH immunocastration effectively suppressed gonadal development in SD rats, which has been confirmed by our previous studies in boars, rats, and rams, using the same GnRH vaccine [10,17]. However, both castration methods exerted profoundly distinct, sex-dependent effects on the gut microbiota ecosystem, which were consistent with previous reports on boars [29] and mice [30,31]. This key difference prompts a deeper investigation into the underlying mechanisms.

Substantial evidence indicated that surgical castration can significantly diminish gut microbial diversity and alter community structure [21,32,33,34,35]. Choi et al. reported decreased gut microbial diversity after ovariectomy [36]. He et al. observed compositional changes in fecal microbiota after surgical castration in boars [29]. Our results align well with these studies, showing reduced microbial diversity in surgical castration rats in both female and male, and altered microbial community structure. In contrast, immunocastration leads to a more gradual and milder decline in hormone levels [37], allowing an adaptation phase for the microbial community to preserve the diversity. We observed that the gut microbial diversity in immunocastrated rats was largely preserved relative to age-matched controls but remained higher than that of surgical castrated rats. In addition, the studies have reported that pro-inflammatory cytokines released systemically can alter gut permeability and directly impact microbial composition [38,39]. The GnRH vaccine works by eliciting a specific humoral immune response [40] to directly or indirectly reshape the gut microbiota. Moreover, the surgical castration is an invasive procedure that inevitably triggers a significant inflammatory response [41], while immunocastration, being a non-invasive treatment, imposes a significantly lesser burden on the gut microbial ecosystem than surgical castration. This divergent gut microbiota was also likely rooted in the fundamental differences in these two interventions.

Beyond the shifts in overall diversity, we observed marked, method-dependent changes at the taxonomic level. The consistent increase, e.g., *Clostridia*_UCG014, *Lactobacillus*, and *Erysipelatoclostridiaceae*_UCG004 in both surgical castration and GnRH immunocastration, whereas a consistent decrease, e.g., *Lachnospiraceae*_UCG006 and *Intestinimonas* in both treatment groups, suggests a common response to the loss of sex hormones. *Lactobacillus*, widely regarded as beneficial bacteria, increased after castration, which has been reported in previous studies [35]. In addition, He et al. also reported that *Lactobacillus* has an obvious gender preference, and there are significant differences in abundance between females and males [29]. Notably, we also identified the opposite responses of *Eubacterium_nodatum* and *Parasutterella* between the two treatments are particularly telling. Given the association of *Eubacterium_nodatum* with beneficial SCFA production [42], its enrichment in the immunocastration group hints at a potentially healthier microbial metabolic output compared to the surgical group, though direct measurement of SCFAs is required to confirm. Moreover, the alterations in nitrogen and glyoxylate metabolism based on the PICRUSt in females point to a fundamental rewiring of microbial energy metabolism post-castration. In males, the changes in siderophore biosynthesis and the TCA cycle suggest a shift in bacterial resource acquisition and energy generation strategies, possibly linked to the dramatic drop in androgens. Together, these findings underscore a potential link between hormone loss and the depletion of beneficial microbes, which may have downstream implications for host physiology.

Furthermore, our results underscored a profound sexual dimorphism in the gut microbiota’s response to both castration procedures. For example, we observed the distinct sets of discriminant genera by surgical castration in females (e.g., *Desulfovibrio*) versus in males (e.g., *Lachnospira*). This divergence between sexes was further magnified when comparing the two castration methods. While the females showed considerable overlap in community structure between immunocastration and surgical castration groups, a marked separation was observed in males. These findings strongly suggested that the gut microbial ecosystem is intrinsically wired differently in each sex, leading to unique successional trajectories upon the loss of gonadal hormones. This aligns with a previous study highlighting the critical interaction between gut-resident bacteria and sex hormones, which influences host development and hormone-mediated disease progression [43]. The relationship is bidirectional, as evidenced by studies showing that the gut microbiota can itself modulate systemic sex hormone levels, such as testosterone [44]. Therefore, the distinct, sex-specific shifts we observed likely arise from a complex dialogue between the host’s endocrine system and the gut microbiome. Unraveling the precise microbial species, genes, and pathways responsible for this cross-talk in future studies will be crucial for developing microbiota-based therapies and deepening our understanding of sexually dimorphic disease. So, a deeper understanding of how immunocastration modulates the gut ecosystem may enable microbiota-targeted interventions that mitigate the adverse consequences of hormonal manipulation.

Despite the comprehensive findings, it is crucial to note the limitations of our study design. Firstly, the surgical and immunocastration groups were likely in different physiological states (e.g., long-term stabilization versus ongoing immune response) at the matched age timepoints. The gut microbiome is highly sensitive to physiological stress and inflammation, and its preservation suggests a more favorable physiological state, a key consideration for animal health. Therefore, the microbiota differences we observed likely reflect a combination of the castration method per se and the distinct post-intervention physiological phase. Secondly, our study design only established correlations between castration methods and microbial shifts, but cannot definitively establish causality, and the microbial metabolic pathways were predicted based on our 16S rRNA gene sequencing data. In addition, while the SD rat model is highly valuable, the translatability of these specific findings to livestock species (e.g., pig, cattle) requires further investigation. Therefore, future research employing metagenomic, metabolomic (e.g., SCFAs, inflammatory markers), and fecal microbiota transplantation approaches will be crucial to validate these predicted functions and establish direct causal links.

## 5. Conclusions

In conclusion, our comprehensive comparison unequivocally demonstrated that GnRH immunocastration and surgical castration were not equivalent in their effects on the gut microbiota, and, strikingly, there are sex-specific influences on the composition and functions of the gut microbiota. These findings illuminated the complex interplay between sex hormones, castration methods, and the gut microbiome. They strongly suggest that the choice of castration technique should carefully consider the potential downstream consequences on the gut microbial ecosystem, with GnRH immunocastration emerging as a promising strategy for maintaining microbial homeostasis. Further study is needed to better understand the cross-talk between sex hormones and the gut microbiota, which could open new avenues of research to identify bacteria that could promote immune regulation and overall animal health.

## Figures and Tables

**Figure 1 animals-15-03512-f001:**
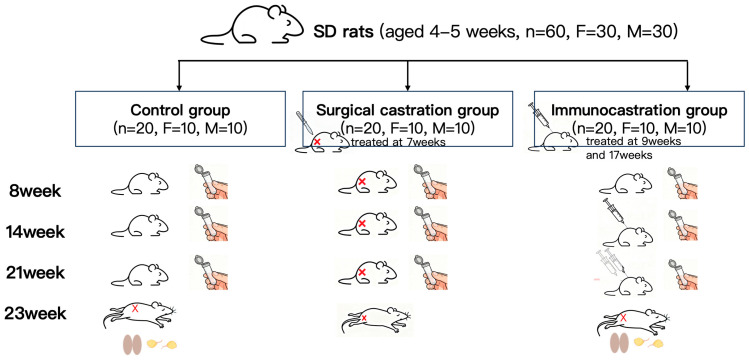
Experimental design. Rats were divided into three groups: control group, surgical castration group, and immunocastration group. Stool samples were collected at 8, 14, and 23 weeks of age. Bilateral ovaries (for females) and testes (for males) were harvested from control and immunocastration groups at 23 weeks of age.

**Figure 2 animals-15-03512-f002:**
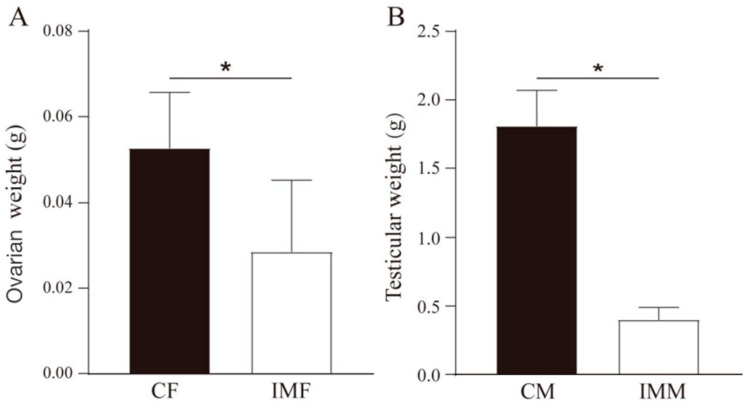
Gonadal weight of SD rats. (**A**) Ovarian weight in female SD rats; (**B**) Testicular weight in male SD rats. * *p* < 0.05 compared to the respective control group. Note: CF, female control group (*n* = 9); CM, male control group (*n* = 10); IMF, GnRH immunocatration for female SD rats (*n* = 9); IMM, GnRH immunocatration for male SD rats (*n* = 10).

**Figure 3 animals-15-03512-f003:**
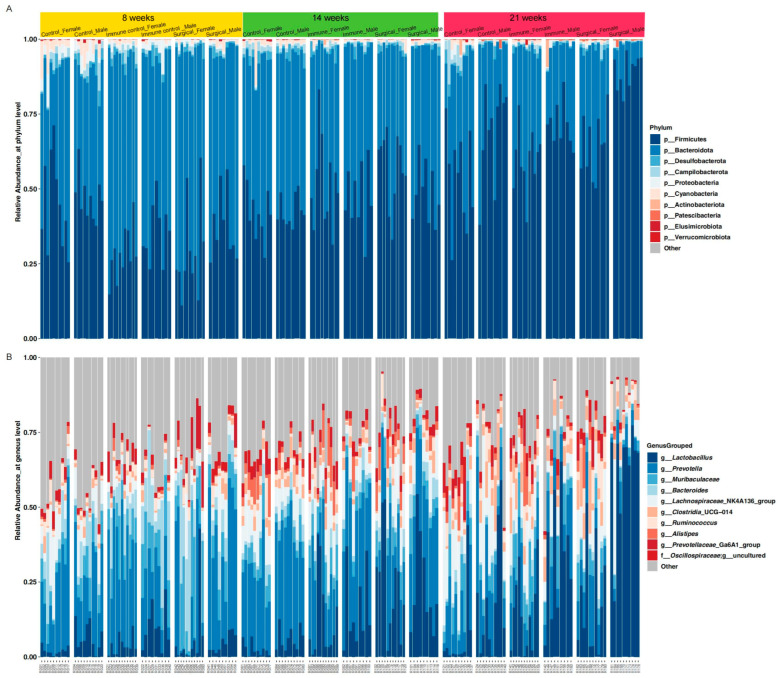
Taxonomic profiles in SD rats across all experimental groups. (**A**) Relative abundance of the top 10 phyla for each individual sample; (**B**) relative abundance of the top 10 bacterial genera for each individual sample.

**Figure 4 animals-15-03512-f004:**
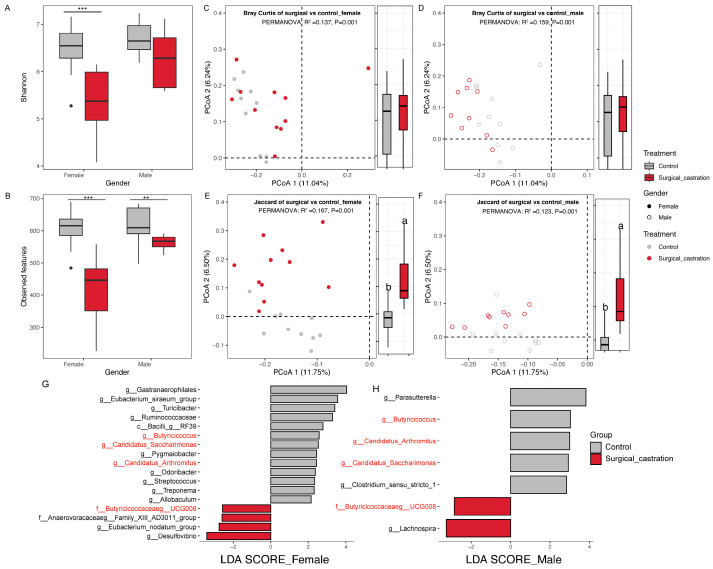
Gut microbiota alterations induced by surgical castration in SD rats. (**A**) Shannon index between surgical castration and control group; (**B**) observed features between surgical castration and control group; (**C**) PCoA based on Bray–Curtis distance for female SD rats; (**D**) PCoA based on Bary-Curtis distance for male SD rats; (**E**) PCoA based on Jaccard distance for female SD rats; (**F**) PCoA based on Jaccard distance for male SD rats; (**G**) Linear discriminant analysis (LDA) of the gut microbiota between surgical castration and control group for female SD rats; (**H**) Linear discriminant analysis (LDA) of the gut microbiota between surgical castration and control group for male SD rats. The black dots in (**A**,**B**) represent outliers. The red labels in (**G**,**H**) indicate microbial taxa significantly vary in both female and male surgical castration groups. For beta diversity, group differences were tested by pairwise PERMANOVA, and boxplots of pairwise distance were on the PCoA (Same as followed). *** *p* < 0.001. ** *p* < 0.001. Different letters denote a significant difference (*p* < 0.05).

**Figure 5 animals-15-03512-f005:**
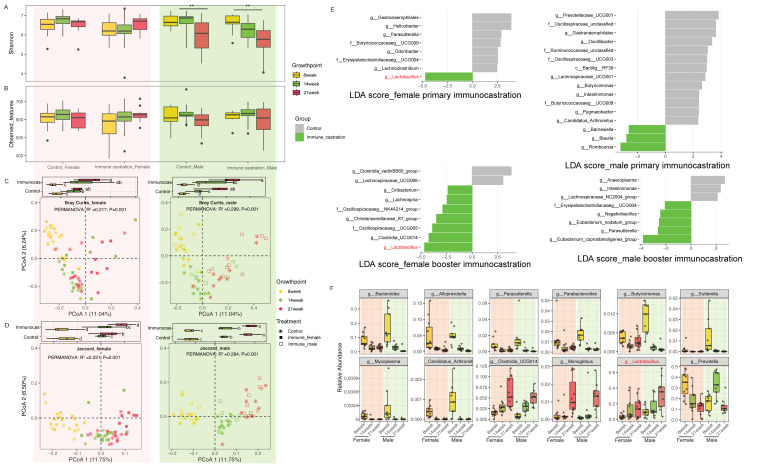
Effects of GnRH immunocastration on gut microbiota in SD rats. (**A**) Shannon index during GnRH immunocastration; (**B**) observed features during GnRH Immunocastration; (**C**) principal coordinate analysis (PCoA) plots showed based on Bray–Curtis distances; (**D**) principal coordinate analysis (PCoA) plots showed based on Jaccard distance; (**E**) changed microbiota after primary and booster immunocastration in SD rats, respectively, which were analyzed by linear discriminant analysis LEfSe; (**F**) relative abundances of significantly changed microbiota in each group. The black dots in (**F**) represent each sample. The red labels in (**E**) represent microbial taxa were enriched in both primary and booster immunocatration groups. ** *p* < 0.001. Different letters denote significant difference (*p* < 0.05); the same letter denotes non-significant difference (*p* > 0.05).

**Figure 6 animals-15-03512-f006:**
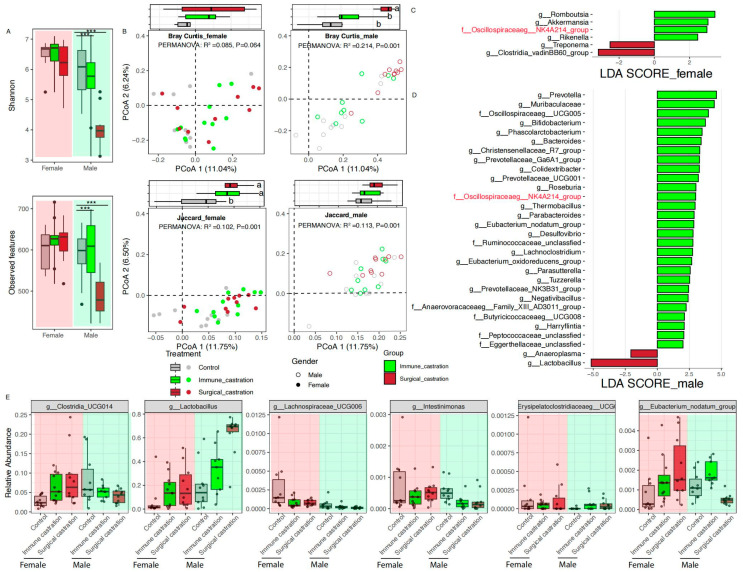
Comparative effects of GnRH immunocastration and surgical castration on SD rats’ gut microbiota. (**A**) Alpha diversity index of Shannon and observed features; (**B**) PCoA plots based on Bray–Curtis and Jaccard distances; (**C**) LEfSe analysis identifying differentially abundant taxa between immunocastration and surgical castration in female SD rats; (**D**) LEfSe analysis identifying differentially abundant taxa between immunocastration and surgical castration in male SD rats; (**E**) Relative abundances of selected microbial biomarker of GnRH immunocastration and surgical castration SD rats. The black dots in (**A**) represent outliers. The red labels in (**C**,**D**) represent microbial taxa were enriched in both female and male immunocatration groups. *** *p* < 0.001. Different letters denote significant difference (*p* < 0.05); the same letter denotes non-significant difference (*p* > 0.05).

**Figure 7 animals-15-03512-f007:**
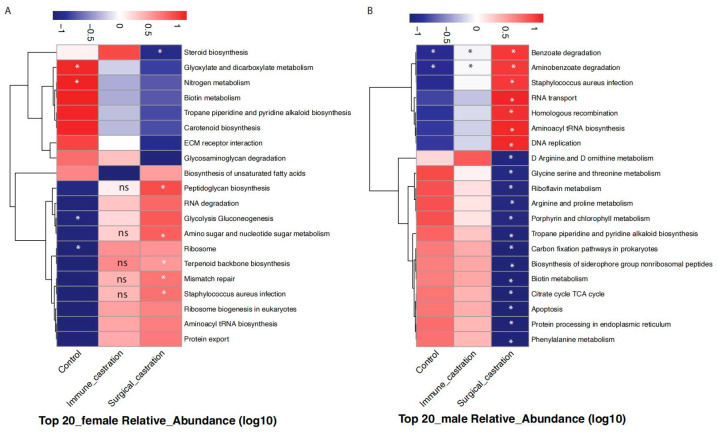
Key predicted microbial metabolic pathways discriminating between immunocastration and surgical castration treatments. (**A**) Random Forest analysis identified the top 20 most important KEGG pathways for distinguishing the two treatments in female SD rats; (**B**) top 20 most important KEGG pathways for distinguishing the two treatments in male SD rats. * indicates a significant difference compared to the other two groups, while ns denotes no significant difference.

## Data Availability

The datasets presented in this study have been deposited in the Genome Sequence Archive at https://ngdc.cncb.ac.cn/gsa/, accessed on 2 December 2025, under accession number CRA031912.

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
