# Peer review of "Distinct Effects of GnRH Immunocastration Versus Surgical Castration on Gut Microbiota"

_animals, 2025, doi:10.3390/ani15243512_

Round 1
Reviewer 1 Report
Comments and Suggestions for Authors
The manuscript entitled "Distinct effects of GnRH immunocastration versus surgical castration on gut microbiota" is a well-designed study, that provides a comprehensive comparison of the impacts of GnRH immunocastration versus surgical castration on the gut microbiota in SD rat model. The key finding that immunocastration preserves microbial diversity to a greater extent than surgical castration has important implications for animal welfare and production practices. The methodologies are sound and state-of-the-art, and the data presentation is generally clear. The manuscript is suitable for publication after minor revisions to address the points outlined below.
- Introduction: The transition from the general role of the gut microbiome to the specific knowledge gap could be slightly smoother.
- While generally understandable, the manuscript would benefit from thorough language editing by a native English speaker or professional editing service to improve fluency and grammatical precision.
- Figure Legends should be more detailed.
- In Result section 3.4: Clarify which specific alpha-diversity metric (Shannon, Observed Features, or both) showed no significant change in females after immunization.
- The discussion could be strengthened by exploring why immunocastration is less disruptive. Elaborate on the potential role of the physical trauma, acute stress, and post-surgical inflammatory response associated with surgery in disrupting gut homeostasis, compared to the more physiological hormone suppression of immunization. Cite relevant literature on surgery/inflammation and the gut microbiome if possible.
- The link to boar taint and human health is excellent. Consider briefly mentioning the potential for microbiome-derived metabolites (e.g., from the altered genera like Lachnospiraceae_UCG006) to influence systemic states, reinforcing the physiological importance of these findings.
- Ensure reference formatting is consistent with the journal's guidelines (e.g., italics for journal names, consistent author name formatting). Some references seem outdated for methodological descriptions (e.g., Ref 27 for PICRUSt instead of Ref 21 for PICRUSt2). Please double-check that the most appropriate and current references are cited for each method.
Author Response
Below, all critique and suggestions provided by reviewers are cited in italics, and our responses are in black.
#Reviewer 1
The manuscript entitled "Distinct effects of GnRH immunocastration versus surgical castration on gut microbiota" is a well-designed study, that provides a comprehensive comparison of the impacts of GnRH immunocastration versus surgical castration on the gut microbiota in SD rat model. The key finding that immunocastration preserves microbial diversity to a greater extent than surgical castration has important implications for animal welfare and production practices. The methodologies are sound and state-of-the-art, and the data presentation is generally clear. The manuscript is suitable for publication after minor revisions to address the points outlined below.
Reply: Thanks a lot for your careful reading and your comments. We have done the following modification.
- Introduction: The transition from the general role of the gut microbiome to the specific knowledge gap could be slightly smoother.
Reply: We thank the reviewer for this excellent suggestion. We agree that the transition could be improved. To address this, we have thoroughly revised the introduction to create a more logical and compelling narrative. Specifically, we have modified the transition “from effect of castration on sex hormone level” to “correlation between sex hormone and gut microbiota” and cite in more studies in this fields. In detail, “Currently, extensive studies have reported the impacts of GnRH immunocastration on reproductive hormone levels and functionality of reproductive organs [16, 17]. While both surgical and GnRH immunocastration ultimately reduce circulating sex hormones levels, their mechanisms of action are fundamentally different [18]. The gut microbiome, a complex community of microorganisms residing in the intestines, plays a pivotal role in host digestion, immunity and metabolism [19].”
- While generally understandable, the manuscript would benefit from thorough language editing by a native English speaker or professional editing service to improve fluency and grammatical precision.
Reply: Thanks a lot for your suggestion. We have asked a native English speaker to improve the language of this manuscript thoroughly, and hope that the revised manuscript meet the requirement of Animals. Changes are tracked throughout the modified manuscript version. Thank you very much!
- Figure Legends should be more detailed.
Reply: We thank the reviewer for this suggestion. We have thoroughly revised all figure legends to provide a more comprehensive and self-contained description, including explanations of the experimental groups, the sample size used in each figure, clear explanation of all symbols (e.g., * P < 0.05). Please see the revised legends for Figures 2, 3, 4, 5, and 6.
- In Result section 3.4: Clarify which specific alpha-diversity metric (Shannon, Observed Features, or both) showed no significant change in females after immunization.
Reply: We have clarified this point in Section 3.4. The sentence now reads: "No significant differences of the Observed Features index were detected either in female or in male SD rats after primary immunocastration and booster immunocastration compared with their corresponding control group (Figure 5B)."
- The discussion could be strengthened by exploring why immunocastration is less disruptive. Elaborate on the potential role of the physical trauma, acute stress, and post-surgical inflammatory response associated with surgery in disrupting gut homeostasis, compared to the more physiological hormone suppression of immunization. Cite relevant literature on surgery/inflammation and the gut microbiome if possible.
Reply: We have expanded the Discussion section. We now suggest that the absence of physical trauma, surgical stress, and the associated systemic inflammatory response in immunocastration likely contributes to its milder impact on the gut ecosystem, citing relevant literature on surgery and microbiome disruption. (Please see the revised Discussion).
- The link to boar taint and human health is excellent. Consider briefly mentioning the potential for microbiome-derived metabolites (e.g., from the altered genera like Lachnospiraceae_UCG006) to influence systemic states, reinforcing the physiological importance of these findings.
Reply: We have strengthened this part in the Discussion include brief, speculative statements on the potential physiological roles of key genera. We now more explicitly discuss how the observed changes in beneficial commensal (Lactobacillus), SCFA-producing bacteria (e.g., Lachnospiraceae_UCG006, which linked to estrogen metabolism) could influence host metabolism and immune function, thereby providing a plausible link between castration method, microbiome composition, and host physiology (e.g., boar taint formation).
- Ensure reference formatting is consistent with the journal's guidelines (e.g., italics for journal names, consistent author name formatting). Some references seem outdated for methodological descriptions (e.g., Ref 27 for PICRUSt instead of Ref 21 for PICRUSt2). Please double-check that the most appropriate and current references are cited for each method.
Reply: We have thoroughly checked the reference list and ensured it now conforms strictly to the journal's formatting guidelines. We have also verified that the most current and appropriate references are cited for all methodologies (e.g., confirming PICRUSt2 is referenced correctly).
Reviewer 2 Report
Comments and Suggestions for Authors
This manuscript compares the effects of GnRH immunocastration and surgical castration on gut microbiota in male and female Sprague Dawley rats using 16S rRNA sequencing. The authors demonstrate that while both methods equally suppress gonadal development, they induce distinct, sex-dependent shifts in microbial composition and predicted functional pathways, with immunocastration appearing less disruptive to the gut ecosystem.
The study addresses an important intersection of animal welfare, endocrinology, and microbiome research. Its strengths include the inclusion of both sexes, integration of microbiome data, clear structure, and fluent writing. However, the manuscript requires substantial revision before publication in a high-impact journal.
Major concerns:
The study remains largely descriptive and lacks mechanistic support. Without measurements of circulating hormones, inflammatory markers, or metabolic indicators, the link between castration type and microbial changes remains inferential. The translational relevance is limited: rat gut microbiota differs markedly from that of other species in composition, fermentation dynamics, and immune-endocrine interactions. Moreover, laboratory rats experience standardized diets and pathogen-free housing, whereas farmed animals face complex nutritional inputs and environmental exposures that profoundly shape microbial communities. In addition, rat immune system has some particularities that may interfere with the microbiota present in the intestinal tract. These interspecies and ecological differences severely limit extrapolation of findings to production animals and should be explicitly acknowledged.
The experimental design has notable limitations. Surgical castration and immunization (establishment of immunocastration is effective after the second shot) occurred at different ages, and no sham-operated control group was included to separate surgical stress from hormonal effects. Statistical analyses lack sufficient detail: multiple comparison corrections, PERMANOVA testing for β-diversity, false discovery rate adjustments, and sequencing depth normalization procedures are inadequately described. Although the authors report the use of Jaccard and Bray–Curtis distances to visualize β-diversity patterns through principal coordinates analysis, these distance metrics alone do not constitute a statistical test of group separation. While PCoA visualizations suggest group separation, formal multivariate analysis is required to statistically validate these differences. Many claims (e.g., "clear separation," "higher alpha diversity") need quantitative support with p-values and effect sizes.
Functional predictions using PICRUSt2 are presented as functional outcomes rather than computational inferences. This limitation requires explicit acknowledgment, and validation through targeted metabolite profiling would substantially strengthen the findings.
To ensure reproducibility, the authors must deposit the raw sequencing data in a public repository such as the NCBI Sequence Read Archive and provide accession numbers within the revised manuscript.
The discussion overstates mechanistic implications. The hypothesis that surgical trauma underlies stronger microbial perturbation is plausible but speculative without physiological measurements or comparisons with a sham-operated control group. Similarly, claims that immunocastration is "less disruptive" or "more welfare-friendly" lack empirical support from behavioral or stress indicators. The use of ether anesthesia is outdated and requires justification or indication of approved alternatives— concerns of an ethical board for the experimental animals’ welfare could have point that. Another concern respects the discussion about the adequacy of rats as experimental models for human and (inferred) also in farm animals. This is highly speculative.
Finally, the references used in the MS should be checked for consistency and cross-checked for their correspondence with numbers in the text. Some of them do not match the text there are referenced to.
Recommendations:
From a technical standpoint, the manuscript is competently executed but does not reach the highest standards of scientific soundness.
To merit publication, the authors must: (1) clarify experimental design and timing; (2) provide transparent statistical analyses with appropriate corrections; (3) perform PERMANOVA to validate β-diversity patterns; (4) moderate mechanistic interpretations; (5) explicitly discuss translational limitations; (6) ideally include hormone and inflammatory marker data; and (7) deposit raw sequencing data in public repositories; (8) revise the reference list and matching references in the text.
Conclusion:
This study offers moderate novelty and clear relevance to animal welfare but lacks the mechanistic depth required for a high-impact publication. With major revisions addressing experimental design, statistical rigor, and interpretive caution, it could make a valuable contribution.
Recommendation: Major revision.

Overall, the English writing is fluent and precise, and the figures are generally clear and informative. Still, there are some misspellings or incongruencies between verbs and noms or adverbs that support the need for a careful language review
Author Response
Below, all critique and suggestions provided by reviewers are cited in italics, and our responses are in black.
This manuscript compares the effects of GnRH immunocastration and surgical castration on gut microbiota in male and female Sprague Dawley rats using 16S rRNA sequencing. The authors demonstrate that while both methods equally suppress gonadal development, they induce distinct, sex-dependent shifts in microbial composition and predicted functional pathways, with immunocastration appearing less disruptive to the gut ecosystem.
The study addresses an important intersection of animal welfare, endocrinology, and microbiome research. Its strengths include the inclusion of both sexes, integration of microbiome data, clear structure, and fluent writing. However, the manuscript requires substantial revision before publication in a high-impact journal.
Major concerns:
- The study remains largely descriptive and lacks mechanistic support. Without measurements of circulating hormones, inflammatory markers, or metabolic indicators, the link between castration type and microbial changes remains inferential. The translational relevance is limited: rat gut microbiota differs markedly from that of other species in composition, fermentation dynamics, and immune-endocrine interactions. Moreover, laboratory rats experience standardized diets and pathogen-free housing, whereas farmed animals face complex nutritional inputs and environmental exposures that profoundly shape microbial communities. In addition, rat immune system has some particularities that may interfere with the microbiota present in the intestinal tract. These interspecies and ecological differences severely limit extrapolation of findings to production animals and should be explicitly acknowledged.“Other than the weight, did you analyse the gonadal histology?”
Reply: We agree with you that providing mechanistic depth and carefully considering the translational relevance are crucial for strengthening our study. We have taken your points seriously and have revised the manuscript accordingly to address these concerns.
Regarding the lack of mechanistic support, we acknowledge that measurements of circulating hormones or inflammatory markers would have significantly strengthened the causal link between castration type and microbial changes. The efficacy of our GnRH immunocastration in drastically suppressing sex hormones (e.g., testosterone, estradiol) has been consistently and robustly demonstrated in our previous studies using the same vaccine in boars, rats, and rams (e.g., Han et al., Theriogenology, 2016; Zeng et al., Anim Reprod Sci, 2002). Therefore, we considered the endocrine outcome as an established premise and focused this work on the previously unexplored consequence—the gut microbiota. However, we fully acknowledge the reviewer's point. In response, we have significantly revised the Discussion section to explicitly state this limitation and clarify that the causal links are inferential. We now elaborate on how the absence of surgical trauma and the distinct pharmacological nature of immunocastration versus the physical removal in surgical castration could lead to differential inflammatory states and physiological stress, thereby shaping the gut microbiota in distinct ways, and cite relevant literature that supports these potential mechanistic pathways.
Furthermore, we fully agree with your critical point on the limited translational relevance of the rat model. We completely agree that direct extrapolation from laboratory rats to production animals must be done with caution due to the well-known differences in gut microbiota composition, immune function, and environmental exposures. In accordance with your suggestion, we have made substantial revisions throughout the manuscript, particularly in the abstract and conclusion sections, to explicitly acknowledge these limitations and reframe the significance of our findings. We emphasize that our findings highlight a previously overlooked aspect that warrants future investigation in target production species, rather than claiming direct applicability.
We show the related comments point by point that the review marked in manuscript.
(1) In abstract Line30, apparently, both sexes were represented. please add information on how many rats were used in each group, and how many were male or female.
Reply: we have added the number of rats in each group marked in red. “Here, sixty Sprague Dawley (SD) rats were randomly allocated into three groups: control (n=20, 10 female and 10 male), surgical castration (n=20, 10 female and 10 male), and GnRH immunocastration groups (n=20, 10 female and 10 male) at 4-5 weeks of age to comparatively investigate the impacts of surgical versus GnRH immunocastration on the gut microbiota”
(2) In introduction line84-86, “this information is conceptually inconsistent. please check and correct”. “castrated mice are not "mature" _ they were”. Try changing into: “Markle et al. demonstrated that the gut microbiota of castrated males shifts toward a female-like composition, indicating that androgens are a major determinant of sex-specific microbial profiles.”
Reply: We thank the reviewer for this comment. We have accepted the reviewer's recommended sentence verbatim and replaced the original text in the Introduction with (now in line 94-97) “Especially, Markle et al demonstrated that the gut microbiota of castrated male mice shifts toward a female-like composition, indicating that androgens are a major determinant of sex-specific microbial profiles”.
(3) “Are rats a good model to farm animals microbiota studies, considering their distinct physiology, nutritional and metabolic differences, and putative differences in host-microbiota-environmental interactions”? And in abstract line52-56, “I have some doubts about the adequacy of the model to farm animals”. And in introduction line91-94 “But not between rats and farm animals...”.
Thank you for your professional comments and suggestions. Carefully consideration, we have revised the text to de-emphasize direct extrapolation to livestock. Specifically, we have moderated the claims of its direct applicability to livestock species in context (e.g. in abstract, in introduction and discussion section). This adjustment allows for a broader interpretation of our findings and enhances its translational value to a wider scientific audience.
- The experimental design has notable limitations. Surgical castration and immunization (establishment of immunocastration is effective after the second shot) occurred at different ages, and no sham-operated control group was included to separate surgical stress from hormonal effects. Statistical analyses lack sufficient detail: multiple comparison corrections, PERMANOVA testing for β-diversity, false discovery rate adjustments, and sequencing depth normalization procedures are inadequately described. Although the authors report the use of Jaccard and Bray–Curtis distances to visualize β-diversity patterns through principal coordinates analysis, these distance metrics alone do not constitute a statistical test of group separation. While PCoA visualizations suggest group separation, formal multivariate analysis is required to statistically validate these differences. Many claims (e.g., "clear separation," "higher alpha diversity") need quantitative support with p-values and effect sizes.
(1) In experimental animal and design section line108, “Please give more information, as there are differences between labs. You need to define the list of pathogens they were free from”
Reply: The specific pathogen-free (SPF) SD rats were purchased from Chengdu Dashuo Experimental Animal Co., Ltd., a licensed commercial vendor. According to the health monitoring report provided by the vendor (in Line 115-118), the animals were confirmed to be free from a standard panel of pathogens (such as Mycoplasma pulmonis, Sendai virus, Pneumonia virus of mice, Salmonella spp., and ecto- and endoparasites.) commonly screened for in rodent facilities in China.
(2) In experimental animal and design section line120 and line123, “please give reference of the GnRH vaccine” and what dose was used?
Reply: We thank the reviewer for this comment. As the reviewer will find, we have already cited the reference for the GnRH vaccine preparation method in the manuscript, and we also added dose of GnRH vaccine. The sentence in line 130-135, which states "The GnRH vaccine was prepared as previously described by Oonk et al. [9, 10, 23]. Briefly, a GnRH juxtaposition dimer using D-lysine to replace glycine at position 6 and ovalbumin linker (TDK-OVA) were dissolved in PBS (pH = 7.4) and fully emulsified with an equal volume of Specol adjuvant. The injection dose (50 mg GnRH peptide equivalent of conjugate) and method were same between primary and booster immuniocastration”.
(3) In 2.3 Sample collection section in line128, “And in the control group, what was the sampling scheme?”, “then the collection moment was different between groups”. “in line131, Was the immune contraceptive vaccine still efffective by this moment”? “In method section, please present a draft for the experimental design, showing also the moment for the interventions and the sampling in each group, start it with the arrival of the animals”. “how were the gonads prepared? particular in the male, the structures weighted included epididymes and blood vessels of the funicle?”, “since the collection momments were different between the groups, the graphs should not be treated as for results in a time-line”, “From the understanding of the descriptions in M&M, I would say that each time identified in the top of the graph represents a different group. If so, ammend the content, in result 3.2 section”, “this sampling moment was not identified in the M&M section in line269-271”
Reply: We sincerely thank the reviewer for these critical questions, which help us to clarify potential ambiguities in our experimental timeline. We apologize for any confusion caused by the initial description.
To clarify, fecal samples were collected from all three groups (Control, Surgical castration, and GnRH immunocastration) at all three specified time points. The time points are defined relative to the interventions for clarity, but samples were collected from every animal simultaneously. We have revised in the text (Line 137-140) to explicitly state this and to align the time points with the age of the animals for easier comparison across groups: “Fresh fecal samples (200mg) from each SD rat in all groups were collected at three time points corresponding to: 8 days post-surgical castration (8 weeks of age), 35 days after primary immunocastration (14 weeks of age), and 27 days after booster immunocastration (21 weeks of age).”
Regarding the efficacy of the GnRH vaccine at the final sampling moment (21 weeks of age, which was 27 days after the booster immunocastration), we confirm that it was still highly effective. The most direct evidence is the significant reduction in gonadal weight observed at the endpoint (23 weeks of age), as presented in original manuscript Figure 1 (now in Figure 2). The Bilateral ovaries and testes were harvested from rats and weighed after removing adhering adipose tissue.
As suggested, we have now included a clear graphical timeline as Figure 1 in the revised manuscript to visually summarize the entire experimental design, including animal arrival and animal number, treatments timepoint (surgical and immunocastration), all sampling time points for all groups, and the endpoint. In addition, the previous figures have been renumbered accordingly.
(4) In statistical analysis line167, “any additional test to explore the effect size?”
Reply: We thank the reviewer for this suggestion. In our study, the effect size for community composition differences was evaluated using the PERMANOVA R² value (reported in the results for beta diversity), and for differential abundant taxa, the LDA score from the LEfSe analysis was used as the standard measure of effect size. We also showed " differentially abundant taxa were detected by the LDA score from LEfSe analysis for boxplot to show the degree to which a taxon's abundance differs between groups such as in Figure 5F, 6E, and Fig S3." We have also ensured that the PERMANOVA R² values are explicitly stated in the figure legends for the PCoA plots.
(5) in Figure 1, “For coherence, in here should be "Ovarian", and add the number of animals in each group”, “This information respect M&M”.
Reply: We have revised the term to "Ovarian weight" for consistency, and the number of animals for each group in this specific analysis has been clearly indicated in figure legend.
(6) the meaning of this sentence in line208 “While the effect of surgical castration on microbial community composition and structure was sex-dependence.” is not clear. Can you reformulate it?
Reply: We have rephrased the sentence as suggested as "However, the effect of surgical castration on microbial community composition and structure was sex-dependent."
(7) in line 245-246, you should not comment on the results section. this may be explained logically, because usually the first vaccine shot fails to obtain complete anullement of the GnHR-gonadotrophin effects.
Reply: We agree that the Results section should present the findings without interpretive commentary. We have removed from the Results section and integrated into the Discussion section to logically explain why the booster immunization was necessary to elicit a more pronounced effect on the gut microbiota.
(8) in line 257-264, the images are too small and when magnifying the page, they became pixelate, give new images with better definition.
Reply: All figures have been regenerated in high resolution and provided as separate lossless PNG files. And each figure was also provided a high-resolution png (lossless) separately in the supplementary materials.
- Functional predictions using PICRUSt2 are presented as functional outcomes rather than computational inferences. This limitation requires explicit acknowledgment, and validation through targeted metabolite profiling would substantially strengthen the findings.
Reply: We have added a sentence in the Discussion section to explicitly state that limitation of the functional predictions inferred from 16S rRNA data.
- To ensure reproducibility, the authors must deposit the raw sequencing data in a public repository such as the NCBI Sequence Read Archive and provide accession numbers within the revised manuscript.
Reply: We have deposited the raw sequencing data in Genome Sequence Archive at https://ngdc.cncb.ac.cn/gsa/ under accession number PRJCA047263. And we added the Data Availability Statement section in the manuscript.
- The discussion overstates mechanistic implications. The hypothesis that surgical trauma underlies stronger microbial perturbation is plausible but speculative without physiological measurements or comparisons with a sham-operated control group. Similarly, claims that immunocastration is "less disruptive" or "more welfare-friendly" lack empirical support from behavioral or stress indicators. The use of ether anesthesia is outdated and requires justification or indication of approved alternatives— concerns of an ethical board for the experimental animals’welfare could have point that. Another concern respects the discussion about the adequacy of rats as experimental models for human and (inferred) also in farm animals. This is highly speculative. the discussion section should be strengthened
Reply: We sincerely thank the reviewer for this constructive and insightful comment. We completely agree that our original discussion overstepped the available data by proposing specific mechanisms without direct experimental evidence. In response, we have substantially toned down the language in the Discussion section to remove speculative claims about the underlying mechanisms (specifically regarding surgical trauma and systemic inflammation).
We acknowledge that ether is an older anesthetic agent. We would like to clarify that its use in this study was specified in and approved by the Animal Ethics and Welfare Committee of Sichuan Agricultural University (Approval No. 20240621), ensuring that all procedures were conducted in accordance with ethical welfare standards. We will utilize contemporary anesthetics like isoflurane in future work.
Finally, the aim of our experiment is to systematically investigate the potential different effects of two distinct methods (surgical castration and immunocastration) on the gut microbiota by using controlled and standardized mammalian models and excluding interfering factors such as diet and environment. We agree with the reviewer that overstating the translational potential of our rat model is speculative. We have therefore revised the text to temper these claims and frame them more appropriately as potential implications and avenues for future research, rather than direct conclusions.
(1) in line250-351, were castrated animals submitted to antimicrobials administration immediatly after susrgery?
Reply: To ensure animal welfare and prevent post-operative infections, we strictly implemented aseptic techniques and local disinfection throughout the entire surgical procedure, while avoiding the use of any systemic antibiotics that would directly affect the gut microbiota.
Finally, the references used in the MS should be checked for consistency and cross-checked for their correspondence with numbers in the text. Some of them do not match the text there are referenced to.
Reply: We have thoroughly checked the reference list and ensured it now conforms strictly to the journal's formatting guidelines.
Recommendations:
From a technical standpoint, the manuscript is competently executed but does not reach the highest standards of scientific soundness.
To merit publication, the authors must: (1) clarify experimental design and timing; (2) provide transparent statistical analyses with appropriate corrections; (3) perform PERMANOVA to validate β-diversity patterns; (4) moderate mechanistic interpretations; (5) explicitly discuss translational limitations; (6) ideally include hormone and inflammatory marker data; and (7) deposit raw sequencing data in public repositories; (8) revise the reference list and matching references in the text.
Conclusion:
This study offers moderate novelty and clear relevance to animal welfare but lacks the mechanistic depth required for a high-impact publication. With major revisions addressing experimental design, statistical rigor, and interpretive caution, it could make a valuable contribution.
Overall, the English writing is fluent and precise, and the figures are generally clear and informative. Still, there are some misspellings or incongruencies between verbs and noms or adverbs that support the need for a careful language review.
We are profoundly grateful for your time and for providing such insightful and constructive feedback on our manuscript. We have found these comments to be exceptionally helpful in strengthening our work. In response to the points raised, we have undertaken a major revision of the manuscript. Specifically, we have (1) clarified the experimental design and timing by adding a detailed schematic and a more precise description in the Methods section; (2) comprehensively revised our statistical analyses to ensure transparency, including stating all tests used, applying appropriate corrections for multiple comparisons, and reporting exact p-values; (3) performed and reported the requested PERMANOVA analysis, which now formally validates the β-diversity patterns observed in our PCoA plots; (4) moderated our mechanistic interpretations throughout the text, particularly in the Abstract and Discussion sections, to more carefully reflect the correlative nature of our findings; (5) added a dedicated paragraph in the Discussion to explicitly acknowledge the translational limitations of our study; (6) deposited the raw sequencing data in a public repository (Genome Sequence Archive, accession number PRJCA047263); and (7) meticulously revised the reference list to ensure accuracy and consistency. Furthermore, the manuscript has undergone professional English language editing to correct any grammatical or stylistic errors. All have been marked in red in the revised manuscript, and we also provide a change-tracked manuscript.
Round 2
Reviewer 2 Report
Comments and Suggestions for Authors
The authors have significantly improved the clarity of the Methods and Results compared with the previous version and have now incorporated PERMANOVA, statistical testing, and better reporting of the pipeline steps, which is appreciated. However, several essential issues remain unresolved, and overall, major revision is still required.
First, the main structural flaw of the experimental design persists. It is unclear whether all groups were sampled at the same age, as the description uses the contraceptive interventions (surgery vs. immunocastration) as the reference point. Notably, the surgery was performed 2 weeks before the first administration of the immunocontraceptive vaccine. The authors' reference anchor is "chronological age" rather than "time since intervention." Throughout this manuscript, the authors compare samples at "8 weeks," "14 weeks," and "21 weeks" as if they represent equivalent biological post-intervention states. Conversely, the authors matched animal age instead of aligning sampling procedures by time since the respective intervention. It should also be considered that the two procedures impair the reproductive axis differently: surgical castration results in abrupt hormone reduction along with acute tissue damage and healing, whereas immunocontraception involves gradual immunological suppression with peak effects occurring only after the booster. Consequently, the microbiome readouts reflect distinct phases of the post-intervention endocrine/inflammatory response in each group and cannot be interpreted as direct "method effects." This confound remains entirely unacknowledged in the manuscript, yet it is central to the interpretation of beta-diversity differences. The manuscript still reads as if differences are attributable to the "method of castration per se," while the data could equally reflect "days post-intervention" physiology. This limitation should be explicitly acknowledged.
Second, the authors did not implement a major requirement in modern microbiome publishing: public deposition of raw sequence data in the SRA (or equivalent) with an accession number. Without this, the microbiome analysis cannot be independently verified.
Third, the mechanistic interpretation is still overstated. Although one sentence now acknowledges that PICRUSt-based pathways are predictions, the Discussion continues to treat pathway differences as real metabolic outcomes. Functional claims should be moderated: the manuscript does not measure SCFAs, inflammatory markers, hormones, or any physiological endpoint except gonad weight at the end. The strong claims regarding "energy metabolism shifts" and "immune and inflammatory pathways" are speculative without supporting biomarker data.
Fourth, the "welfare-friendly" conclusion remains unsupported. There are no behavioral stress indicators, pain scores, post-operative analgesia descriptions, or corticosterone measurements. Gut microbiota differences alone do not constitute welfare evidence. The welfare claim should be softened or removed unless supported by appropriate behavioral/physiological readouts.
Fifth, the Discussion still implicitly extrapolates rat data to production animals. The SD rat model is highly controlled, with SPF housing and standard chow, and does not reflect pig gastrointestinal ecology, diet complexity, or immune–endocrine kinetics. Please add a paragraph explicitly acknowledging that SD rat microbiota is not representative of pig microbiota and that the translational value is mechanistic only.
Finally, although reporting has improved, the current version still reads largely as descriptive. Given the strong sex-dependent signals, the authors could improve interpretation by explicitly discussing sex × treatment × time interactions rather than simply listing genera that differ.
In sum, this manuscript remains publishable in principle but needs one further round of revision focused not on adding more analyses, but on interpretive accuracy, transparent data availability, and explicit limitation framing. The study adds incremental knowledge to the microbiota–sex hormone literature, but at this stage, I cannot recommend acceptance without the above adjustments.
Comments on the Quality of English LanguageOverall, the English writing is fluent and precise, and the figures are generally clear and informative. Still, there are some misspellings or incongruencies between verbs and noms or adverbs that support the need for a careful language review
Author Response
A point-by-point response to the reviewers' comments is attached, detailing all the modifications made. The key improvements in this revised version include:
The authors have significantly improved the clarity of the Methods and Results compared with the previous version and have now incorporated PERMANOVA, statistical testing, and better reporting of the pipeline steps, which is appreciated. However, several essential issues remain unresolved, and overall, major revision is still required.
First, the main structural flaw of the experimental design persists. It is unclear whether all groups were sampled at the same age, as the description uses the contraceptive interventions (surgery vs. immunocastration) as the reference point. Notably, the surgery was performed 2 weeks before the first administration of the immunocontraceptive vaccine. The authors' reference anchor is "chronological age" rather than "time since intervention." Throughout this manuscript, the authors compare samples at "8 weeks," "14 weeks," and "21 weeks" as if they represent equivalent biological post-intervention states. Conversely, the authors matched animal age instead of aligning sampling procedures by time since the respective intervention. It should also be considered that the two procedures impair the reproductive axis differently: surgical castration results in abrupt hormone reduction along with acute tissue damage and healing, whereas immunocontraception involves gradual immunological suppression with peak effects occurring only after the booster. Consequently, the microbiome readouts reflect distinct phases of the post-intervention endocrine/inflammatory response in each group and cannot be interpreted as direct "method effects." This confound remains entirely unacknowledged in the manuscript, yet it is central to the interpretation of beta-diversity differences. The manuscript still reads as if differences are attributable to the "method of castration per se," while the data could equally reflect "days post-intervention" physiology. This limitation should be explicitly acknowledged.
Reply: We sincerely thank the reviewer for this insightful comment regarding the experimental design, that the surgical castration and immunocastration operate through fundamentally different mechanisms and kinetics. So, we used chronological age as the sampling anchor to align with the developmental stages of the animals and to minimize acute procedural stress.
Specifically, in Results 3, which focuses on the effects of surgical castration on gut microbiota, both the control and surgical castration groups were sampled at 8 weeks of age. Surgical castration treatment was performed at 7 weeks of age, allowing for a one-week recovery period to reduce the impact of procedural stress prior to sampling. In Results 4, where we investigated the effects of immunocastration and its dynamic changes on gut microbiota, the control and immunocastration groups were sampled at 14 weeks (5 weeks after the primary immunocastration, immediately before the booster) and 21 weeks (4 weeks after the booster immunocastration). This sampling schedule was designed to avoid the acute stress of vaccination and to capture the established effects of the immunization. The timelines for all interventions and sampling points are detailed in Section 2.2 and 2.3 of the Materials and Methods and illustrated in Figure 1.
We fully acknowledge, however, the inherent limitation of this design, particularly in Results 5 and 6, where we directly compared the surgical and immunocastration groups at the 21-week time point. We selected the 21-week time point for the direct comparison the impact on the gut microbiota between surgical castration and immunocastration based on our prior physiological data. “The hormone levels such as testosterone, LH, and FSH were markedly decreased at 2wpv and then remained low or nondetectable at 4 wpv after immunocastration compared with intact males (reference 10 in manuscript). We therefore reasoned that this time point would allow for a more valid comparison of the longer-term, stable effects of each method on the gut microbiome, minimizing the confounding influence of acute hormonal fluctuations or active immune responses. Although a control group was also included at this same age, the two treatment groups were indeed likely in different physiological phases at the moment of sampling—the surgical group being in a long-term, stable state of hormone ablation, and the immunocastration group potentially still under the influence of an active immune response. Consequently, as the reviewer rightly points out, the observed microbial differences likely reflect a combination of the castration method itself and the distinct post-intervention physiological state.
In recognition of this critical point, we have added the following statement to the Discussion section to explicitly acknowledge this limitation:
"It is important to note that a key limitation of our study design. Given the distinct modes of action -- acute surgical insult with immediate hormone ablation versus gradual immunologically-mediated suppression -- the surgical and immunocastration groups were likely in different physiological states (e.g., long-term stabilization versus ongoing immune response) at the matched age timepoints. Therefore, the microbiota differences we observed likely reflect a combination of the castration method per se and the distinct post-intervention physiological phase. Future studies designed with a unified 'time-post-intervention' anchor are needed to disentangle these effects."
Second, the authors did not implement a major requirement in modern microbiome publishing: public deposition of raw sequence data in the SRA (or equivalent) with an accession number. Without this, the microbiome analysis cannot be independently verified.
Reply: We fully agree that public data deposition is essential for reproducibility. We have deposited all raw sequencing data in the Genome Sequence Archive (GSA, https://ngdc.cncb.ac.cn/gsa/) at the National Genomics Data Center, China National Center for Bioinformation on 02/10/2025. The specific accession number for our dataset is CRA031912 or PRJCA047263. As is standard practice for this database, the data are currently under a private status until the associated manuscript is accepted for publication. Upon acceptance, the data will be immediately made public. We have now explicitly stated this accession number in the "Data Availability" section of the revised manuscript.
We thank the reviewer for highlighting this critical oversight, and we believe this now fulfills the requirement for full data transparency.
Third, the mechanistic interpretation is still overstated. Although one sentence now acknowledges that PICRUSt-based pathways are predictions, the Discussion continues to treat pathway differences as real metabolic outcomes. Functional claims should be moderated: the manuscript does not measure SCFAs, inflammatory markers, hormones, or any physiological endpoint except gonad weight at the end. The strong claims regarding "energy metabolism shifts" and "immune and inflammatory pathways" are speculative without supporting biomarker data.
Reply: We sincerely thank the reviewer. We have carefully considered all points raised and have made significant revisions to address the specific concern regarding the overstatement of mechanistic interpretations.
We meticulously revised the text in both the Results and Discussion sections to ensure that all descriptions of functional pathways are clearly framed as predictions. The key changes are as follows:
We removed the standalone results section titled "3.6 Sex-Specific Functional Alterations..." in Results Section. The content comparing the predicted functional profiles of the two castration methods has now been integrated into the end of Section 3.5 ("Comparative Analysis of Gut Microbiota Alterations..."). This change prevents these predictive findings from being presented with undue weight as a primary result. and now we introduce this part with the phrase: “At last, we used a computational tool, PICRUSt (Phylogenetic Investigation of Communities by Reconstruction of Unobserved States) [27], to explore metabolic potentials of...”. We added qualifiers such as "predicted", "potential" when mentions of microbial functions and pathways.
We have significantly toned down the interpretation of the functional predictions, and added a dedicated paragraph in the Discussion Section to explicitly acknowledge this limitation. For example, in line 421-427 “Secondly, our study design only established correlations between castration methods and microbial shifts but cannot definitively establish causality. And our study did not directly measure microbial gene expression, metabolite levels (e.g., SCFAs, inflamma-tory markers), or host circulating hormones beyond the initial gonad weight confirmation. The PICRUSt we used, predicted the microbial metabolic pathways based on our16S rRNA gene sequencing data. Therefore, these functional interpretations remain specu-lative and highlight the need for future validation”.
Fourth, the "welfare-friendly" conclusion remains unsupported. There are no behavioral stress indicators, pain scores, post-operative analgesia descriptions, or corticosterone measurements. Gut microbiota differences alone do not constitute welfare evidence. The welfare claim should be softened or removed unless supported by appropriate behavioral/physiological readouts.
Reply: We sincerely thank the reviewer for raising this critical point. We fully agree that animal welfare is a multifaceted concept requiring direct assessment through behavioral, physiological, and clinical measures. Our study was not designed with these specific welfare endpoints. It was an overstatement to directly conclude that GnRH immunocastration is "welfare-friendly" based solely on gut microbiota data. So, we remove all direct and unsupported claims regarding "welfare" and have reframed our conclusions to focus on the potential implications of our findings for gut ecosystem stability, which is an important component of overall animal health. The key changes are as follows:
In the Simple Summary, the concluding sentence has been revised to: “This suggests it is a potentially less stressful alternative that better preserves a stable gut environment, which is an important aspect of animal health”.
In the Abstract, the final sentence was revised to: “and provided insights for developing humane and effective approaches to animal population control.”
A statement was added in the Discussion: “Despite the comprehensive findings, it is crucial to note that the limitations of our study design. Firstly, the surgical and immunocastration groups were likely in different physiological states (e.g., long-term stabilization versus ongoing immune response) at the matched age timepoints. The gut microbiome is highly sensitive to physiological stress and inflammation, and its preservation suggests a more favorable physiological state, a key consideration for animal health. Therefore, the microbiota differences we observed likely reflect a combination of the castration method per se and the distinct post-intervention physiological phase. Secondly, our study design only established correlations between castration methods and microbial shifts but cannot definitively establish causality, and the microbial metabolic pathways were predicted based on our16S rRNA gene sequencing data. In addition, while the SD rat model is highly valuable, the translatability of these specific findings to livestock species (e.g., pig, cattle) requires further investigation. Therefore, future research employing metagenomic, metabolomic (e.g., SCFAs, inflammatory markers), and fecal microbiota transplantation approaches will be crucial to validate these predicted functions and establish direct causal links. ”
Fifth, the Discussion still implicitly extrapolates rat data to production animals. The SD rat model is highly controlled, with SPF housing and standard chow, and does not reflect pig gastrointestinal ecology, diet complexity, or immune–endocrine kinetics. Please add a paragraph explicitly acknowledging that SD rat microbiota is not representative of pig microbiota and that the translational value is mechanistic only.
Reply: We completely agree that the Sprague-Dawley rats under specific pathogen-free (SPF) conditions and a standard chow, represents a highly controlled environment that is distinct from the complex gastrointestinal ecology, diverse diets, and different immune-endocrine kinetics of production animals like pigs. We acknowledge that the rodent gut microbiota is not directly representative of the porcine microbiota. The translational value of our study lies primarily in elucidating the fundamental mechanistic principles of how different castration methods can distinctly impact the gut ecosystem in a sex-dependent manner, rather than in providing direct, one-to-one predictions for livestock. To address this concern directly, we have added a new, dedicated paragraph in the Discussion section to explicitly acknowledge this limitation and clarify the scope of our conclusions.
Finally, although reporting has improved, the current version still reads largely as descriptive. Given the strong sex-dependent signals, the authors could improve interpretation by explicitly discussing sex × treatment × time interactions rather than simply listing genera that differ.
Reply: We have thoroughly revised the Discussion section to explicitly frame our interpretations, and we add a paragraph to improve interpretation between the sex difference and gut microbiota.
“Furthermore, our results underscored a profound sexual dimorphism in the gut microbiota's response to both castration procedures. For example, we observed the distinct sets of discriminant genera by surgical castration in females (e.g., Desulfovibrio) versus in males (e.g., Lachnospira). This divergence between sexes was further magnified when comparing the two castration methods. While the females showed considerable overlap in community structure between immunocastration and surgical castration groups, a marked separation was observed in males. These finding strongly suggested that the gut microbial ecosystem is intrinsically wired differently in each sex, leading to unique successional trajectories upon the loss of gonadal hormones. This aligns with previous study highlighting the critical interaction between gut-resident bacteria and sex hormones, which influences host development and hormone-mediated disease progression [43]. The relationship is bidirectional, as evidenced by studies showing that the gut microbiota can itself modulate systemic sex hormone levels, such as testosterone [44]. Therefore, the distinct, sex-specific shifts we observed, likely arise from a complex dialogue between host’s endocrine system and the gut microbiome. Unraveling the precise microbial species, genes and pathways responsible for this cross-talk in future study will be crucial for developing microbiota-based therapies and deepening our understanding of sexually dimorphic disease. So, a deeper understanding of how immunocastration modulates the gut ecosystem may enable microbiota-targeted interventions that mitigate adverse consequences of hormonal manipulation. ”
In sum, this manuscript remains publishable in principle but needs one further round of revision focused not on adding more analyses, but on interpretive accuracy, transparent data availability, and explicit limitation framing. The study adds incremental knowledge to the microbiota–sex hormone literature, but at this stage, I cannot recommend acceptance without the above adjustments.
We sincerely thank the reviewer for your professional suggestions to improve our manuscript. We have fully addressed all the points raised, with this round of revisions focused precisely on enhancing interpretive accuracy, transparent data availability, and explicit limitation framing, as recommended.